# Gene disruption by structural mutations drives selection in US rice breeding over the last century

Justin N. Vaughn[1,2]*, Walid Korani[2], Joshua C. Stein[3], Jeremy D. Edwards[4], Daniel G. Peterson[5], Sheron A. Simpson[1], Ramey C. Youngblood[5], Jane Grimwood[6], Kapeel Chougule[3], Doreen H. Ware[3,7], Anna M. McClung[4], Brian E. Scheffler[1]*

1 USDA-ARS, Genomics and Bioinformatics Research Unit, Stoneville, Mississippi, United States of America, 2 University of Georgia, Athens, Institute of Plant Breeding, Genetics, and Genomics, Athens, Georgia, United States of America, 3 Cold Spring Harbor Laboratory, Cold Springs Harbor, New York, United States of America, 4 USDA-ARS, Dale Bumpers National Rice Research Center, Stuttgart, Arkansas, United States of America, 5 Mississippi State University, Institute for Genomics, Biocomputing & Biotechnology, Starkville, Mississippi, United States of America, 6 Hudson-Alpha Institute for Biotechnology, Huntsville, Alabama, United States of America, 7 USDA-ARS, Robert W. Holley Center for Agriculture and Health, Ithaca, New York, United States of America

* justin.vaughn@usda.gov (JNV); brian.scheffler@usda.gov (BES)

**Data Availability Statement:** If not in supporting information, relevant data files and software are located at https://www.ncbi.nlm.nih.gov/bioproject/PRJNA603026 and https://github.com/USDA-ARS-

## Abstract

The genetic basis of general plant vigor is of major interest to food producers, yet the trait is recalcitrant to genetic mapping because of the number of loci involved, their small effects, and linkage. Observations of heterosis in many crops suggests that recessive, malfunctioning versions of genes are a major cause of poor performance, yet we have little information on the mutational spectrum underlying these disruptions. To address this question, we generated a long-read assembly of a tropical *japonica* rice (*Oryza sativa*) variety, Carolina Gold, which allowed us to identify structural mutations (>50 bp) and orient them with respect to their ancestral state using the outgroup, *Oryza glaberrima*. Supporting prior work, we find substantial genome expansion in the *sativa* branch. While transposable elements (TEs) account for the largest share of size variation, the majority of events are not directly TE-mediated. Tandem duplications are the most common source of insertions and are highly enriched among 50-200bp mutations. To explore the relative impact of various mutational classes on crop fitness, we then track these structural events over the last century of US rice improvement using 101 resequenced varieties. Within this material, a pattern of temporary hybridization between medium and long-grain varieties was followed by recent divergence. During this long-term selection, structural mutations that impact gene exons have been removed at a greater rate than intronic indels and single-nucleotide mutations. These results support the use of *ab initio* estimates of mutational burden, based on structural data, as an orthogonal predictor in genomic selection.

GBRU/US_rice, respectively. Chromosome-scale scaffolds for Carolina Gold assembly can be found at https://de.cyverse.org/dl/d/2C3CF540-2962-4BC2-8131-6CE8AA4FA4FE/oryza_carolina-toplevel-20180831.fa.gz.

**Funding:** JNV, JDE, DHW, AMM, BES are supported by Agricultural Research Service (USDA) projects: ARS projects: 6066-21310-005-00-D, 6066-21310-005-23-S, 6066-21310-005-25-S, 8062-2100-044, and 6028-21000-011-00D. See https://www.ars.usda.gov/ for more details. The funders had no role in study design, data collection and analysis, decision to publish, or preparation of the manuscript.

**Competing interests:** The authors have declared that no competing interests exist.

## Author summary

Some crop varieties have superior performance across years and environments. In hybrids, harmful mutations in one parent are masked by the ancestral alleles in the other parent, resulting in increased vigor. Unfortunately, these mutations are very difficult to identify precisely because, individually, they only have a small effect. In this study, we use long-read sequencing to characterize the entire mutational spectrum between two rice varieties. We then track these mutations through the last century of rice breeding. We show that large structural mutations in exons are selected against at a greater rate than any other mutational class. These findings illuminate the nature of deleterious alleles and will guide attempts to predict variety vigor based solely on genomic information.

## Introduction

Though the details vary substantially, most major crops have undergone a dramatic reduction in genetic diversity relative to their wild progenitors due to domestication and breeding [1]. In the case of rice, this reduction is compounded by self-pollination. These early events have had a substantial impact on the contemporary population's mutational load or its "cost of domestication". Strong selection on a few key domestication loci and long-term reductions in population size resulted in the expansion and even fixation of many mildly deleterious alleles [2]. Rice was one of the earliest models used to examine and demonstrate this cost at the sequence level [3].

A central task of modern breeding is to purge these deleterious alleles [4]. Historically, this purging has been accomplished indirectly through extensive crossing and phenotypic selection. Genomic data offers a complementary (or perhaps wholly alternative) way of explicitly identifying the alleles most likely to be deleterious [5]. In effect, with the knowledge of all deleterious mutations in hand, breeders could dramatically accelerate genetic gain through targeted crosses and much larger populations afforded by marker-based breeding value assessment of progeny [4].

While the theoretical basis for the cost of domestication is strong, few studies have investigated how this cost is manifested in long-term, realistic agronomic settings. In addition, large structural mutations (SVs) are likely to have the most dramatic phenotypic effects [6]. Indeed, they account for a disproportionate number of discovered causal variants [1,4]. Gene dysregulation is analogous to (and often driven by) presence/absence variation resulting from structural mutations; in maize, this dysregulation is correlated with allele frequency and is a predicator of seed weight [7]. Yet, structural events have been recalcitrant to the short-read sequencing deluge. Recent work in rice using 3,000 re-sequenced rice genomes shows a clear depletion in coding-sequence indels, a hallmark of inbreeding and selection [8]. Though this work used an extensive SV-calling pipeline, false-positive and false-negative rates were still in excess of 10% and often much greater for tandem duplications and inversions [8,9]. More importantly, it remains unclear if the segregating structural variants are present because they are neutral or because deleterious alleles have not yet been fully purged.

The US rice industry traces its commercial production along the southeast coast to the 1700s. Since *Oryza* spp is not indigenous to the USA, rice varieties from around the world were imported and evaluated for production potential through the 1920s. Currently, approximately 80% of the 1 M ha of rice grown in the USA are planted in the Mid-South region along the Mississippi River and Gulf Coast and utilize tropical *japonica* germplasm (https://www.nass.usda.gov/). Early breeding efforts in the USA focused on a relatively narrow genepool of

tropical *japonica* germplasm that possessed the combination of agronomic and grain quality traits desired by the domestic industry. An analysis of 24 cultivars developed over a 48 yr time period in Texas revealed an average decrease in days to heading (> 0.2 cm yr-1), decreased plant height (> 1.0 cm yr-1), and an increase in whole milling yield (0.06% yr-1) and grain yield (24 to 42 kg yr-1) [10]. In the late 1950s, as the rice milling and processing industry developed, criteria for physicochemical, cooking, and processing quality traits were established to guide breeders in the development of cultivars having superior quality [11]. Similarly, breeding for resistance to disease has also been a major goal. For example, several *Pi* blast resistance genes have been deployed over six decades in the USA [12].

From this US-bred material, we generated a long-read assembly of a foundational tropical *japonica* variety, Carolina Gold Select (Pi 636345) (shortened to "CarGold", in the following text and figures). This assembly was used to generate a high-confidence set of SVs spanning the genome. In addition, we resequenced 166 varieties representing USA rice breeding efforts over the last century. Of these, we focused on 101 varieties that have been advanced in a consistent environment and have had documented gene flow based on robust pedigree information. Together this subset allowed us to characterize the full mutational spectrum across a well-defined time course of breeding and selection in the USA.

## Results and discussion

### *O. sativa* genome size continued to increase after the temperate/tropical split due to a small set of retrotransposons

Large structural mutations have resulted in genome expansion in the *Oryza sativa* lineage [13]. TE activity–a major driver–appears to have diminished although there is evidence for continued transpositional activity at low frequency in contemporary breeding material [14]. Given the repetitive nature of TEs, their activity can be very difficult to assess using short-read sequencing technology. Long-reads, which can span the critical ~12 kb threshold of full-length retrotransposons, have proven central to characterizing TE events at sequence-level resolution. In order to identify TE events and additional complex SVs, we employed PacBio long-read technology to generate a *de novo* assembly of a foundational US rice variety, Carolina Gold ("CarGold"). In addition to its significance to US rice breeding, CarGold is representative of the major tropical branch of the *O. sativa* japonica subpopulation.

The CarGold long-read assembly resulted in 208 contigs with an N50 of 12.88 Mb (**S1 Table**). PacBio contigs were scaffolded into pseudomolecules using the Nipponbare temperate *japonica* reference (IRGSP-1.0.59), 97.4% of which was covered by the CarGold assembly (**S4 Fig**). In addition to high coverage, the CarGold assembly shows excellent contiguity with Nipponbare, even for centromeric content, and nearly identical core gene content (**S2 Table**). The centromere of chromosome 6 is one exception: it appears to contain three major inversions, although we did not pursue confirmation of these given that the internal content appears to be contiguous and they represent gene poor regions outside the scope of the study. In both general metrics and fine-scale accuracy relative to an Indica outgroup, this PacBio assembly appears to be superior to a recently released assembly of the same variety that used a hybrid Nanopore/Illumina approach [15] and offers an interesting comparison of the two methodologies (**S12 Fig**). Most notably, the Nanopore assembly appears to be prone to collapsing repeats into false insertions. Interestingly, while collinearity with an Indica variety genome supports internal structure of chromosome 6 inversion, the recent Carolina Gold Select Nanopore assembly supports the large-scale Nipponbare configuration (see above and **S12 Fig**). Whether or not this is a convergent scaffolding error remains to be determined but such an error is

plausible given the inter-contiguous nature of the break in the otherwise accurate CarGold PacBio assembly.

To characterize >50bp SVs, CarGold, Nipponbare, and *O. glaberrima* chromosomes were aligned. Using the *O. glaberrima* outgroup [16], the ancestral state of an indel between *japonica* subtypes could be inferred (**Fig 1**A and [13,17]). As expected, the majority of events were in the outgroup relative to *O. sativa*. Because they cannot be inferred, such outgroup events were removed in the following analyses. All inferred events were compared with SVs identified using CarGold PacBio reads directly aligned to Nipponbare reference and called with pbsv (https://github.com/PacificBiosciences/pbsv, v2.2), currently the best SV caller for long-read data [18]. We expected far fewer events to be present in our set since we require stringent border alignments and outgroup alignment. Indeed, whereas pbsv calls 18,238 SVs, we called 5,571 SVs (**S4 File**). 82% of the SVs we inferred overlapped pbsv SVs. Of those that did not, 92% involved insertion of 5% or more of the total SV. Manual curation revealed that the insertion was generally present in aligned reads but either the SV was in a hemizygous state or a deletion occurring in conjunction with the insertion (see below) had apparently disrupted pbsv's ability to call the SV.

Most events we inferred were, as expected for a double-strand break repair process [19], a mixture of inserted and deleted bases, although generally one of the two appear to dominate an event and insertion-like events are much more frequent (**Fig 1**B). In addition to being more frequent, the insertions involve more bases on average (**Fig 1**C). The length profiles are consistent across the two varieties (**Fig 1**C). Summing across all events indicates a net gain of ~6 Mb in both lineages. As our inference methodology was conservative, this value represents a lower bound on the net gain.

The contribution of particular length-classes was not uniform (**Fig 1**D). While we observed some very large events (>40 kb), these have very little impact on the cumulative length of inserted sequence (~13 Mb). The most impactful length class was centered around 11 kb, wherein two well-defined "step-changes" were observed (**Fig 1**D). Using a full-length TE database generated specifically for this study (see Materials and Methods), we annotated the TEs within all insertions and observed a clear enrichment of two Gypsy retrotransposon families within the step-changes (**Fig 1**D). Together these two families account for ~1.5 Mb (or 11%) of inserted sequence (**S3 File**). An additional set of Copia elements, ~ 6k bp in length, accounts for another 835kb (or 6%).

## Most insertions are tandem duplications generated via patch-repair

While TEs, as a whole, account for 28% of the total length of added DNA, they only directly account for 9% of the total events. Though limited by short-read methodologies, prior work in rice identified that tandem duplications accounted for at least half of all new insertions >10 bp [17]. These events did not appear to result from replication slippage, but DNA repair of adjacent nick sites as indicated in [17,20]. The CarGold assembly allowed us to extend prior results, which were limited to primarily <100bp events, to those SVs described above that are 10 to 100-fold longer.

We calculated a *d* metric for CarGold vs Nipponbare indels (**Fig 2**), which is a numerical description of the ambiguity in alignment between ancestral and derived states [21]. Direct integration of ectopic DNA has a $d = 0$. Events such as replication slippage, which should depend on small annealing sites, have a *d* greater than the length of the indel. In perfect tandem duplication, *d* equals the exact length of the indel. Our results are complementary to prior observations [17]. Importantly, deletions do not conform to the 1:1 relationship seen for insertions, indicating that tandem duplication is much more common than tandem removal, the latter likely occurring through unequal crossing over. This asymmetry also supports our ability to accurately infer indel ancestral state using *O. glaberrima* as an outgroup.

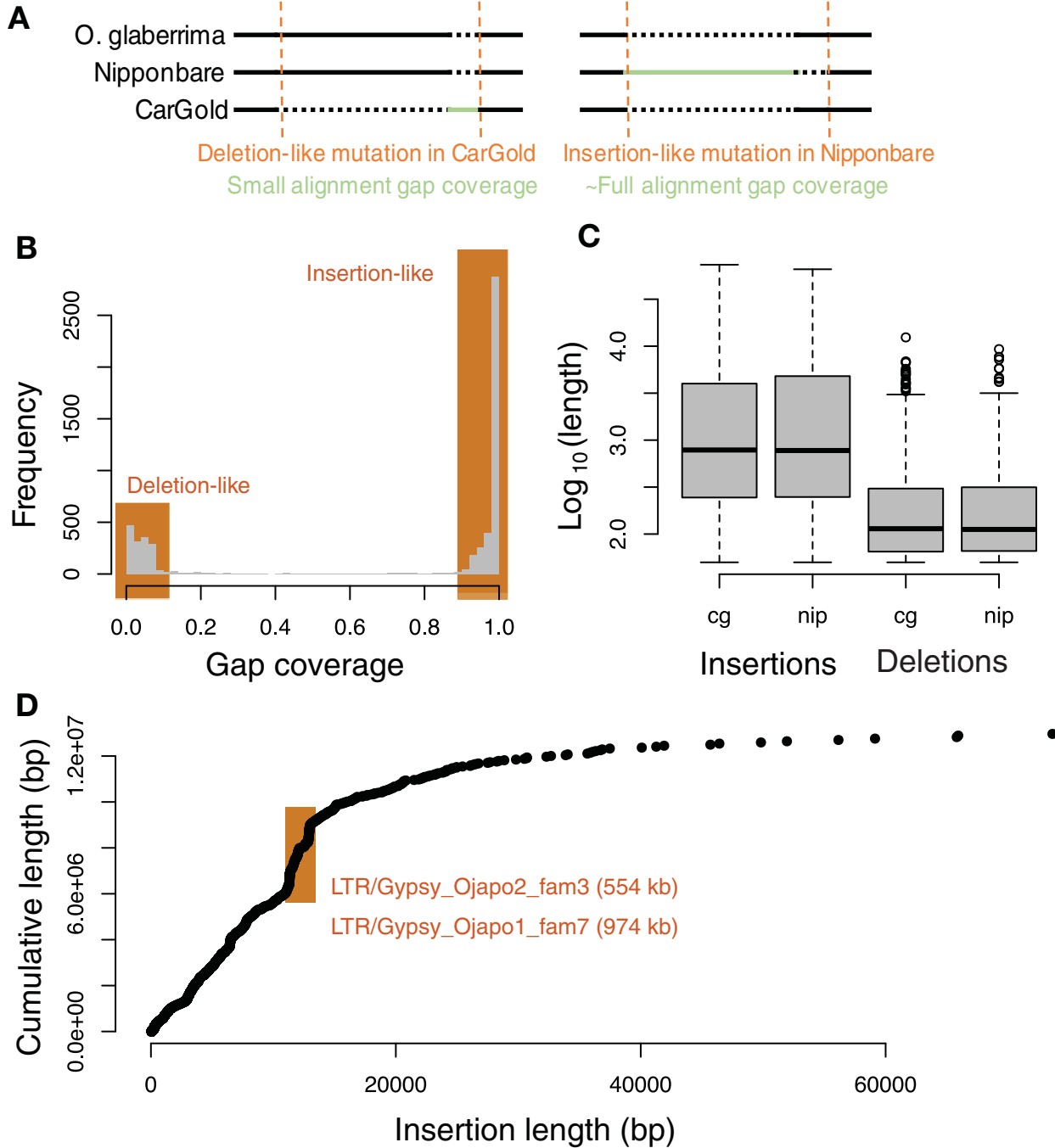

**Fig 1. Insertions and deletions between temperate and tropical japonica references.** A) Schematic illustrating the insertion/deletion orientation method for indel characterization. Any mutation found only in *O. glaberrima* is ambiguous and ignored. B) Distribution of gap coverage values across all events analyzed. As indicated, insertions and deletions are defined as mutations with a gap coverage of >95% or <5%, respectively. C) Boxplot depicting the log-transformed size distribution of events broken out by type and the variety–CarGold (cg) or Nipponbare (nip)–in which it was derived. D) Scatterplot showing each insertion, sorted by length, and its impact on cumulative length of all insertions. TEs contributing to rapid changes in total inserted sequence are indicated by orange window.

Because we examined much longer mutations, we were able to identify an upper limit to the tandem duplication mechanism. As indicated in **Fig 2**, the relative proportion of tandem duplications declines as events exceed 125 bp. This proportion starts at 70% of events and falls

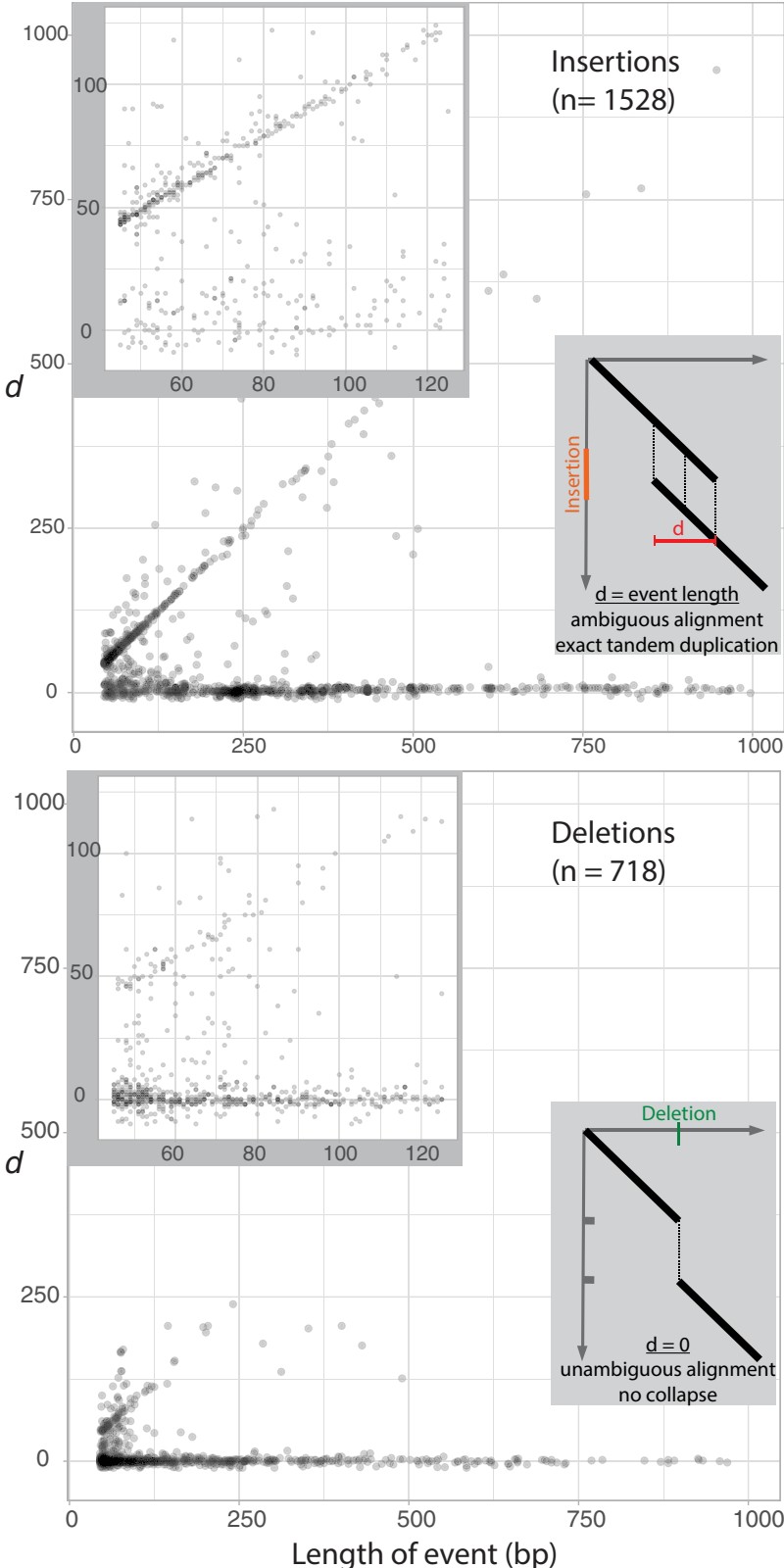

**Fig 2. Tandem bias in insertions versus deletions between Nipponbare and CarGold.** Indel size is plotted against the *d* metric for each insertion and deletion satisfying our alignment and inference criteria. Upper corner insets show

only indels <125bp. Lower corner insets give two examples of how $d$ is calculated; alignments between ancestral and derived (indicated by red or green) sequence are depicted as simplified dotplots.

to 5% by 200 bp. Interestingly, the proportion of tandem duplications appears to remain at 5% out to the longest values used in the analysis (15 kb). Indeed, we observed perfect tandem duplications in excess of 1 kb, with the longest being 4.9 kb.

Across both varieties, we examined 5,571 >50bp mutations. As described in the Introduction, the presence of many of these events are expected to be a consequence of the domestication bottleneck. Given increased population size and improved measurement accuracy of yield under modern breeding, a signature of selection pressure to purge these possibly deleterious alleles should be detectable. To interrogate this relationship in an agricultural context, we turned to the US rice germplasm collection, which has representative sampling of released varieties spanning the last century of rice breeding in the US.

## Admixed introductions are followed by targeted breeding efforts for long and medium grain markets

One-hundred and sixty-six rice varieties developed or used in US rice breeding were sequenced as Illumina 100–150 bp, paired-end reads (**S1 File**). For population analysis, single nucleotide variations (SNPs) were called relative to the Nipponbare reference. This set of ~11.6M SNPs was cross-filtered with SNP data derived from ~3k other rice varieties [22], resulting in ~3.9M intersecting SNPs.

For examining the population structure in the US sample, we further filtered full SNPs by retaining those SNPs with low pairwise LD between one another [23]. Thirty randomly sampled representatives from each rice subpopulation found in pre-existing data on ~3k rice varieties [22] were used as training populations to infer the proportional origin of each US variety (**Fig 3**A).

US rice varieties with substantial temperate *japonica* content are primarily derived from California breeding programs. MidSouth varieties are generally tropical *japonica*, but a substantial fraction exhibit admixture between the two *japonica* types. Grain type appears to be the underlying trait that differentiates admixed MidSouth varieties from purely tropical varieties (**Fig 3**B). While >8% temperate admixture generally are associated with medium-grain varieties, there are clearly exceptions: in some cases ~50% admixture can exhibit long-grain type, whereas <2% can still have medium-grain type. Though varieties tend to be genetically distinct by grain type, admixtures are clearly present (**Fig 3**B), indicating gene flow between the subpopulations as a result of breeding efforts.

We examined population structure of varieties over the course of the last century by ordering each Midsouthern variety based on year of release (**Fig 3**B). Red dots indicate a variety's release date relative to its genetic relatedness to all other varieties. Generally, varieties are more closely related to varieties released within the same time period, as expected for breeding populations. The pattern of relatedness also reflects general market history. Initially, admix populations derived from foreign sources were introduced in the early 1900s from within which selections were made that were adapted to the southern USA. The majority of varieties that exhibit unusual seed type given their population assignments appear during this time period. Controlled crosses were implemented in 1929 with a focus on long grain development for the next 25 years. With this market established, there was an emphasis to also develop a medium grain market with several released in the 1950s. However, after 1960 over 70% of the new releases were long grains. Thus, over time, the total population has diverged through breeding and selection and resultant gene pools targeted to long and medium grain market classes.

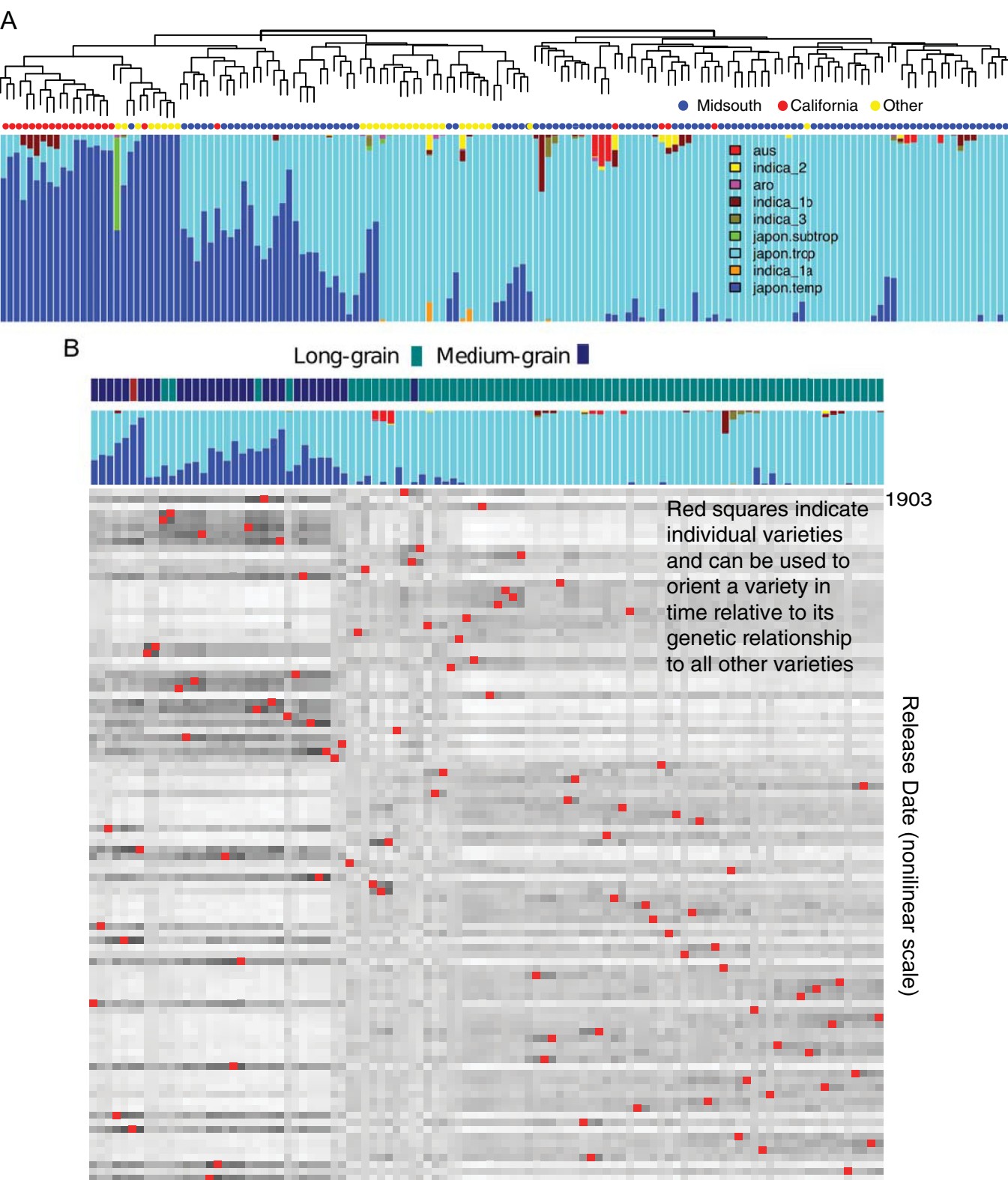

**Fig 3. The genetic structure of US rice breeding varieties.** A) The population structure of sequenced US germplasm samples in this study. The dendrogram represents hierarchical clustering based on a kinship matrix derived from the low LD marker set (See Materials and Methods). Bar plots, based on the same markers, indicate the proportion of genetic content of each individual that can be assigned to a set of known rice subpopulations. Colored dots represent the

general location/program from which material was derived. B). Heatmap of the centered-identity-by-state (IBS) between each variety in the analysis and all other varieties, both along the column and row axis. Columns are clustered based on similarity. Rows are ordered explicitly by date of release. The red squares connect a variety's clustering position with its release-date position. Seed type is indicated along the top, clustering axis. "Short grain" shown as maroon.

## Structural mutations are being purged from the germplasm

This historical collection of varieties allowed us to further examine the impact of different mutational classes in a long-term breeding context. Indeed, selection against SVs is supported by our initial observation of one SV per ~424kb of exonic space versus ~222kb of intronic space (described below). Though biased TE distribution may be a factor, TE-mediated events are the minority and this ~2-fold bias almost certainly reflects the historical purging of SVs in exons. Comparable results were recently observed in *O. sativa* using short-read SV inference [8]. Yet it remains unknown if the SVs that we do observe escaped negative selection because they are neutral or if selection has been constrained by linkage and narrow genetic bottlenecks related to domestication (see Introduction). To examine this question, we explored changes in assorted mutational classes in the Midsouthern population.

Using release-date as a surrogate for agronomic fitness, we estimated the effects of the large structural variants discovered above as well as a random subset of SNPs, for which we were able to define an ancestral state using *O. glaberrima* as above. Alleles were called based on read contiguity across these pre-ascertained sites (see Materials and Methods). Read coverage across all varieties was not significantly correlated with reference allele calls ($p$-val = 0.08) and a linear regression model indicates that, if there was a significant relationship, it would only account for 1.7% of our variance in genome-wide genotyping (S10 Fig). We additionally used 207x CarGold short-read sequencing data to independently assess false-negative allele calls among those SVs that were segregating in Midsouthern varieties. Assuming the SVs defined above are all true, no CarGold allele should be called as the Nipponbare reference. We down-sampled reads randomly to reflect the interquartile coverages across the US varieties: 22x, 31x, and 40x, single copy coverage. A false-negative rate of 10% across all events (S10 Fig) is rivaled only by the ability to call small deletions using short-read data in prior studies [8,9]. Indeed, manual curation suggests "miscalls" were generally the result of heterozygous-appearing loci (see above), potentially resulting from tandem duplication. Since our approach is referenced biased, these SVs will inevitably get called as Nipponbare alleles. Thus, our effective false-negative rate is probably lower than 10%. The frequency of SVs across all Midsouthern varieties was highly correlated with the additional diverse germplasm sequenced in this study (**Figs 3** and **S13**, and **S7 File**), and, as expected for a foundational line, CarGold-derived alleles were on average more common in Midsouthern material than Nipponbare-derived alleles.

All allele calls (SNPs and SVs) relative to the ancestral state were linearly regressed on release-date to give the rate of change across all samples [24]. This methodology is comparable to that used previously [25], although, because we are comparing groups of variants spread relatively evenly through the genome (S6 Fig), population structure should not be a confounding factor. Unfortunately, it is difficult to assess variants that have a major impact on gene structure from an ancestral perspective because the reference, Nipponbare, was used to determine that gene structure. For example, ancestral deletions that remove a gene in Nipponbare will be missed because the resultant affected region simply lacks the gene to be annotated [26]. To that end, we focused only on mutations in which the Nipponbare reference matches the ancestral state and so by inference originated in the CarGold lineage. Within this set, we focused on six major mutational classes: exonic, intronic, and intergenic SVs and SNPs that introduced stop-codons, non-synonymous amino acids, or have no effect on protein coding (synonymous) (**Fig 4**).

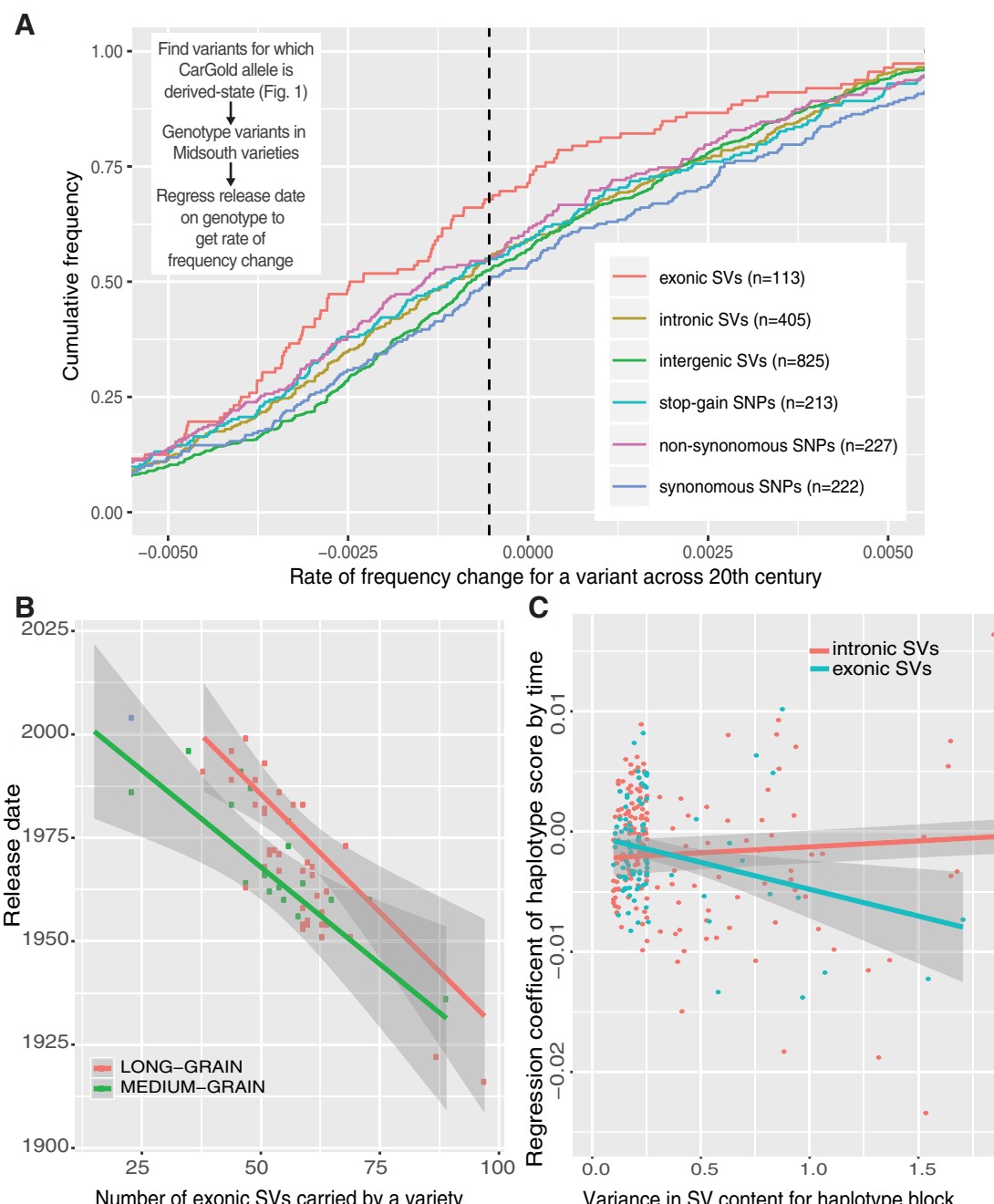

**Fig 4. The rate of change in functional variant classes across a century of plant breeding.** Rate estimates are reported only for mutations that have occurred in CarGold, not Nipponbare (see Text). A) Cumulative frequency of rates for each variant class. The farther a class is shifted to the left of 0, the more rapidly it has declined on average. B) Plot and linear fit of release date versus the total count of alleles that disrupt an exon in each variety for each grain type. C) Relationship between change in haplotypes scored by number of exonic indels and the variance in this score (see Fig 5). The y-axis represents, for each LD block, the regression coefficient of a linear model between haplotype score and the release date of the variety containing that haplotype. The x-axis represents the variance for those haplotype scores. In B and C, the shading around the regression line represents the 95% confidence intervals in combined intercept and slope estimates.

All median rates for each class are shifted to the left of synonymous SNPs. Given that these are CarGold-derived variants, the negative shift for synonymous SNPs, indicated by dotted line in **Fig 4**A, likely reflects a general selection against unadapted CarGold haplotypes.

Though the signal is small in some cases, the median values for each class are roughly ordered based on the expected impact on open reading frame disruption (**Fig 4**A). Least significant difference test (implemented in "agricolae" library in R with p.adj = "fdr") supports the following grouping of effect classes: Synonymous SNPs and intergenic SVs shared similar profile, particularly with regard to the most negative rates. Intronic SVs and both non-synonymous and stop-gain SNPs are effectively equivalent with median values shifted to the left of low effect mutations. Exonic SVs are substantially more negatively shifted than any other group. Within this group, we did not observe any bias with regard to length or deletion/insertion status (**S5 Fig**), supporting the expectation that any event >50 bp has an equivalent impact on gene disruption.

SVs had comparable density across chromosomes (**S6 Fig**), and results were essentially unaffected when chromosomes with the most divergent patterns (Chr 6 or 11) were removed from the analysis. If intergenic read mapping is less accurate than exonic mapping, a negative relationship between a variety's year-of-release and its genetic distance to the reference could skew results: allele calls would be falsely biased toward the derived mutation in intergenic regions of newer varieties. In fact, we observe the opposite: a marginal positive relationship between year-of-release and distance to reference (**S9 Fig**), indicating that the signal of selection against gene disrupting SVs is, if anything, slightly diluted. There was also no discernable relationship between a mutation type, its rate of change, and its false-negative genotyping rate described above (**S10 Fig**). Lastly, in the case of true selection, rates based on linear regression will be strongest for variants that have a minor allele frequency of 0.5 across the entire sample. Derived allele frequencies were different between mutational classes, as expected for alleles that have a different impact in general (**S11 Fig**). Exonic SVs exhibit relative enrichment between 0.15 and 0.3. Again, this indicates that exonic rate estimate is biased toward weaker, not stronger, values.

Many of the varieties sequenced in this study had available pedigree information. Twenty-seven parent/progeny trios could be formed from this data. These trios segregated for an average of ~18% of SVs used in this study (**S14 Fig**). SV and SNP mis-calls were correlated for most trios (**S15 Fig**), although exceptions appear to be enriched among the "Bonnet" lineages. Extensive miscalling suggests that pedigree information is incorrect. We excluded Palmyra and CI9701 trios given their high level of miscalls relative to segregating variants, leaving 25 trios for further analysis. Using SVs derived in the CarGold lineage (as above), we filtered any SV that did not segregate in at least 10 trios and any SV for which one of its two alleles was never inherited–suggesting common parent miscall. We then assessed inheritance bias relative to the derived allele of all SVs. As above, classes are based on positioning relative to coding regions. Though statistical power is much reduced using trios, the results support conclusions based on the year-of-release regressions across the entire population (**S16 Fig**). Moreover, the trio results further exclude population structure as a confounder of year-of-release analysis.

Based on these results, we expected to be able to partially predict a variety's year-of-release from the sum of exonic SVs that it contained. We observed that the relationship between reduced exonic SVs and later release date holds for both grain-types (**Fig 4**B), in spite of the fact that these events are all CarGold-derived due to reasons discussed above. In the long-grain material, the relationship is heavily driven by the exonic SV content of the oldest varieties and may not be useful in contemporary selection. That said, these events represent only a fraction (20–30% by our rough estimate) of the exonic SVs segregating at >10% in this population. Given the strength of signal we observe ($R^2$ = 0.24, *p*-val = 1.63e-06), we predict that the addition of those data would substantially improve prediction accuracy (see Conclusions). These predictions do not include the impact of other mutational classes as well.

Estimates of the effects of any variant will be complicated by linkage: a locus containing a beneficial and deleterious allele in positive phase can be effectively invisible to selection,

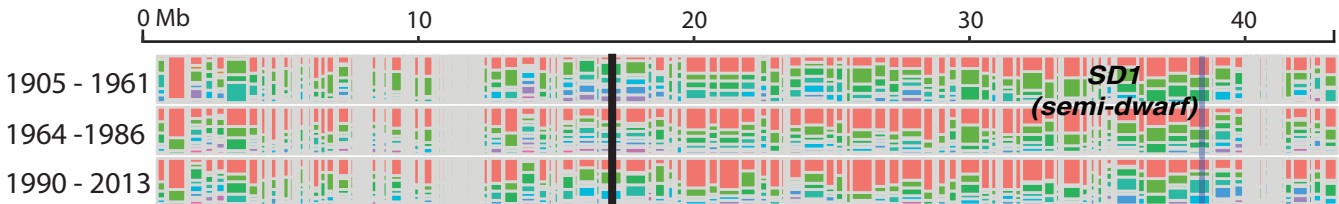

**Fig 5. The select-ome of US rice breeding.** Chromosome 1 is shown as representative. The chromosome is divided into LD blocks and haplotypes within those blocks are colored based on red being the most frequent across the three time periods defined on far left. Semi-dwarf locus, a known target of selection, is labeled. Sparse regions represent very small LD blocks. All chromosomes are plotted in **S8 Fig**. See https://gbru-ars.shinyapps.io/HaploStrata/ for fully interactive plots.

assuming these alleles are of equivalent effect. This relationship between linkage and selection should be apparent from our data as well. To test this (and to facilitate locus-specific selection analysis below), we defined LD blocks for the combined MidSouth population using a down-sampled set of ~60k SNPs (**Fig 5**). (Increased coverage had only a minor impact on refining LD blocks further, see Materials and Methods.) We counted the number of derived exonic structural mutations present in each haplotype within a given LD block. We estimated a regression coefficient between this score of each haplotype and the year-of-release of the variety in which it was present. Intronic SVs were used as a conservative "neutral" control, given that they are positionally correlated and show some bias (**Fig 4**A). Our hypothesis was, for exonic SVs, that the LD blocks with the largest variance in haplotype score would exhibit the most negative regression coefficients. Exonic SVs do exhibit this relationship relative to intronic SVs (**Fig 4**C). Due to ascertainment and probably selection, the majority of our data contains only 1 exonic SV per LD block and thus we have few exonic data-points with substantial variance. Still, the regression coefficient for exonic indels is statistically significant (p-val < 0.01) as indicated by 95% confidence intervals (**Fig 4**C), supporting the idea that haplotype context has an impact on an individual allele's ultimate trajectory [24,27].

The genes disrupted by structural mutations were slightly biased toward functioning in biotic and abiotic stress response (**S3 Table**). Inclusion of genes harboring strongly declining intronic mutations strengthened this signal (**S4 Table**). Still, the genes implicated in this analysis only account for ~5% of disrupted genes, suggesting that the bulk of the effect is driven by a general disruption of metabolic function [7]. There was a slight enrichment in overlap with loci found to effect heterosis in rice [28], though the significance of the bias (18 observed versus 10 expected) is difficult to assess in this context.

## 3% of LD blocks exhibit sweep-like changes in frequency

The difference in rates of allele frequency change between functional classes of mutations indicates that genetic drift in this rice population is not substantial enough to mask the signature of selection of certain loci. While weakly deleterious alleles appear to have a cumulative effect across the genome, highly adaptive mutations can occur against this backdrop and sweep through a population. We can track haplotypes through time (see above) and identify those with aberrant changes expected from such a sweep. Still, the identification of selective sweeps in unreplicated populations is notoriously problematic [29]. Positive controls can be useful in establishing a realistic threshold for selection [30,31]. Starting in the late 1960s, the semi-dwarf phenotype was introduced into Asian and then US breeding programs, often increasing yields in modern agronomic environments by 50% or more [32]. Though multiple alleles exist, the major source in our population is the Taichung Native 1 allele derived from an *indica* source [33].

For all haplotype blocks in our sample, we partitioned varieties into 3 time periods: early (1905–1961), middle (1964–1986), and late (1990–2013). The *sd1* block indicates a rapid increase in two nearly identical haplogroups by 23% from middle to late eras (**Fig 5**). Using this value as a threshold, 3% of LD blocks exhibit an equivalent or greater change in single haplotypes from one era to another. We also overlaid known agronomic genes (n = 46, **S6 File**) and examined the overlap with frequency change. Though numerous genes overlapped, including the *Pi-ta* locus that determines resistance to a number of pathotypes of the rice blast fungus (*Magnaporthe oryzae*) [34], this proportion did not deviate substantially from random expectation. Thus, many putatively selected regions have no obvious target of selection (**S5 File**). To facilitate further exploration of these regions, an interactive tool complementary to **Fig 5** is available at https://gbru-ars.shinyapps.io/HaploStrata/

The LD block sizes in our population represent the diversity in the population, either generated by recombination or by introduction of novel haplotypes. Strong selection on rare or *de novo* mutations can produce LD blocks in later time-periods that are much larger than those of earlier generations [35]. Such a pattern would not be immediately evident in **Fig 5**. Barring heightened selection for recombination, the rate of change in alleles and the ratio of LD block lengths in early versus late generations should be highly correlated.

By evaluating pairwise identity across early and late generations (or "HScan Index", see Materials and Methods), we did note a general shift toward longer linkage blocks (**S2** and **S7 Figs**), reflecting the population divergence observed above (**Fig 3**) and mild depletions in diversity. The distribution also has a prominent tail, likely reflecting selected outlier loci. The differences in block lengths are associated with selected loci, as defined at the Δ23% level above, exhibiting 11% longer LD in the late versus early intervals (**S3 Fig**). Interestingly, the *sd1* locus exhibits reduction in the HScan index. This change is due to the loci originally being homogenous in early types; as *sd1* began to sweep into but not completely through the population, it has actually increased diversity. Only one other locus (on chromosome 7, **S7 Fig**) had such reduced diversity in the early era and it too shows substantial increase in diversity through time.

Soft sweeps occur when an allele appears in multiple haplotypes prior to an increase in selection efficiency [35]. These events can be very difficult to detect because the change in any one of the containing haplotypes is quite small. Only one of the 367 selected LD blocks possess multiple selected haplotypes. Though this percentage is almost certainly a low estimate of soft sweeps given the sensitivity afforded by our sample size, the full scope of this research supports a model in which advantageous mutations of the same locus will generally only exist in a small number of haplotypes fit enough to rise in frequency.

## Conclusions

Current methodologies for genomic selection use phenotypic and genotypic data from a training population to predict the yield of varieties that have little or no phenotypic information [36]. The accuracy of the prediction is highly correlated with the relatedness between the training and test populations, and predictions can be made from a small number of markers [37]. Such findings fit well with an infinitesimal model of yield: selection models based on a single training population lack generality because an unrelated test is comprised of very distinct sets of linked alleles. The effect of these individual alleles on yield can never be realistically estimated in a QTL-mapping context, but it may be possible to gauge their effect if evaluated as a class of mutations. Variety performance could then be predicted as a sum of these effects or as a composite model with phenotypic and genotypic data [5]. In this study, we have attempted to define which mutational classes would be the most relevant to such an approach by looking

at selection across a broad agronomic environment, namely, Midsouthern US varieties in the 20$^{th}$ century. Sequence conservation scores, such as GERP, have been used previously to generate *de novo* breeding values using SNPs [5]. Our results indicate that while SNPs are relevant, predictive power could be substantially improved by considering SVs, particularly those that disrupt exons. That said, we did not observe an obvious way to weigh different exonic SVs in terms of phenotypic consequences: length was not a factor (**S5 Fig**), suggesting that an additive GERP score would be inappropriate. Though not addressed here due to sample size, phylogenetic distribution of homologs may be a more useful correlate of effect size under the assumption that disruptive mutations in broadly distributed orthologs will have greater effects. Still, results using a simple 0/1 weighing scheme are encouraging (**Fig 4**B and 4C). Importantly, they are based solely on one reference genome annotation and SVs derived from a single, alternative assembly. Critical to the advancement of such *ab initio* prediction will be 1) the availability of enough high-quality *de novo* assemblies to capture common SVs in breeding germplasm and 2) the computational tools to annotate genes within this pan-genomic context.

## Materials and methods

### Carolina Gold PacBio sequencing and assembly

High molecular weight DNA was extracted from young leaves using prior protocol with minor modifications [38]. Essentially, young leaves, that had been flash frozen, and kept frozen at -80C, were ground to a fine powder in a frozen mortar with liquid N2 followed by very gentle extraction in CTAB buffer (that included proteinase K, PVP-40 and beta-mercaptoethanol) for 1hr at 50C. After centrifugation, the supernatant was gently extracted twice with 24:1 chloroform:iso-amyl alcohol. The upper phase was adjusted to 1/10$^{th}$ volume with 3M KAc, gently mixed, and DNA precipitated with iso-propanol. DNA was collected by centrifugation, washed with 70% Etoh, air dried for 20 min and dissolved thoroughly in 1x Tris-EDTA at room temperature. Size was validated by pulsed field electrophoresis.

Libraries were prepared using PacBio SMRTbell Template Prep Kit 1.0, PacBio SMRTbell Damage Repair Kit, and prepared for sequencing using PacBio DNA/Polymerase Binding Kit P6 V2. Sequencing was performed on the PacBio RSII using PacBio DNA Sequencing Reagent 4.0 v2 and PacBio SMRT Cell 8Pac V3. All protocols used were PacBio recommended protocols.

Raw PacBio reads (n = 3,765,107; ~70x coverage) were assembled into 209 contigs, using `Canu` (v1.5) [39], and polished with `Quiver` (`smrtlink` v5.0.1 suite, now at *github.com/ PacificBiosciences/GenomicConsensus*). PacBio raw reads were further polished with `pilon` v1.22 [40] using "10X Genomics" linked-reads aligned by `Longranger` v2.1.6 (*github.com/ 10XGenomics/longranger*). The original primary assembly consisted of 209 contigs. One contig (tig74) was determined to be a false chimeric assembly and was split into tig74a and tig74b. Repeat masking was performed as part of the `MAKER-P` pipeline [41] using custom repeat libraries, PReDa_121015_short.fasta (DNA) and TE_protein_db_121015_short_header.fasta (protein) [42,43].

### Whole genome alignments

PacBio contigs of CarGold were assigned to Nipponbare, a temperate *japonica* variety, chromosomes (*Osativa_204_v7.0.softmasked.fa*). Exclusive pairwise relationships were determined based on the sum coverage of CarGold scaffold by collinear Nipponbare sequence. Any scaffold with a sum coverage of <65% was removed. CarGold scaffolds and their respective Nipponbare chromosomes were combined with the appropriate *Oryza glaberrima* chromosome [16] and aligned in a chromosome-wise manner using *progressiveMauve* [44] (build-date-Feb-

25-2015) with default parameters. Both Nipponbare and *O. glaberrima* genomes were generated using numerous methodologies–BAC Sanger, WGS Sanger, 454, among other—as distinct from PacBio long reads; this diversity of methodologies should reduce or eliminate the possibility of convergent errors due to technical artifacts.

## Insertion/deletion inference

Indels were identified from the whole genome alignments above and were polarized relative to the outgroup, *Oryza glaberrima*, using a custom program (indelInference.pl). Columns in the whole chromosome alignments that involved >50 consecutive gaps (in any sequence) were extracted along with +/- 50 bp of flanking sequence. Gaps were analyzed further if the left and right flanking regions aligned with >90% columns being identical. If Nipponbare and CarGold shared 95% identity in the gapped region, the SV was not considered further. Alternatively, if there was variation between Nipponbare and CarGold and one matched *O. glaberrima* with >95% identity, then the event was inferred to have occurred in the non-matching sequence. This approach captured the biological reality that mutation events creating long (>50bp) SVs rarely involve only insertion or deletion of DNA but a combination of both. To that end, we also characterized the degree to which with each mutation represents a net gain or loss of DNA. The length of the entire gapped region was divided by the length of novel sequence introduced in the gap such that values approaching 0 are, in effect, deletions and values approaching 1 are insertions. (A small number of SVs with gap values between 0.49 and 0.51 were removed after manual curation indicated these "perfectly balanced" indels represent unwarranted gap openings.)

## Reference sets for TE content

An initial set of full-length TEs from the *japonica* subpopulation was retrieved from RiTE [43] (3 March 2018). This set was searched against both the Nipponbare and CarGold assemblies using *nucmer* (version 3.1) with–maxmatch. Alignments were filtered such that a full-length TE had to align to 97% of its length at a 95% identity threshold. The matching genomic region was extracted if it was not already assigned to another TE meeting the above criteria. The Nipponbare and CarGold TE sequences were combined. This combined set was searched against itself using *nucmer* with—maxmatch flag. Coordinate files were then filtered such that sequences with a reciprocal overlap of 90% were paired. *Mcl* (version 14) was used to consolidate pairs into clusters. All sequences within a cluster were aligned with mafft (version 7.307). A consensus sequence was generated using a custom program (majorityRuleConsensus.pl), such that any alignment column with A, C, G, T, or–(gap) was called based on the most frequent variant. If the gap is the most frequent, then the column is removed from the final consensus. This pipeline resulted in 7,624 TE consensus sequences (i.e. families)(**S2 File**).

## Characterizing insertion and deletion types using *d* metric

The *d* metric was calculated for insertions and deletions as described previously [17,21]. In brief, ancestral and derived sequences including the inserted/deleted sequence and flanking sequence (equal in length to the indel) were aligned using *BLAST* (v.2.6.0+) [45]: 'blastn -gapopen 2 -gapextend 4 -dust no'. The two major alignments were identified and the top-alignment end position was subtracted from the bottom-alignment start position, relative to the longest of the two aligned sequences (**Fig 2**)

### Short-read DNA sequencing of US germplasm samples

DNA was extracted using https://www.protocols.io/view/seed-sterilization-and-tissue-disruption-for-optim-bpsemnbe. All sequencing libraries where generated using the Illumina TruSeq PCR-free reaction. Accession libraries were sequenced on different instruments: 24 libraries on Illumina HiSeq2500 in paired-end mode with 101 cycles, 70 libraries on Illumina HiSeqX in paired-end mode with 151 cycles, and 72 libraries on Illumina HiSeq3000 in paired-end mode with 151 cycles. Raw Illumina reads and *bbtools khist.sh* (jgi.doe.gov/data-and-tools/bbtools/) were used to assess the single-copy k-mer count for each accession, based on the main peak in the resultant kmer spectrum (**S1 Fig**). Haploid genome size was also estimated as part of this analysis (**S1 File**).

### SNP-calling, merging, and pruning

Raw reads were trimmed using *trimmomatic* [46]. Reads were then aligned to *Osativa_204_v7.0.softmasked.fa* using *bwa-mem*, version 0.7.17 [47]. HaplotypeCaller from the GATK suite, version 4.0.8.1, was used to call SNPs and small indels [48]. Each sample was used to generate a GVCF file (-T HaplotypeCaller—genotyping_mode DISCOVERY—emitRefConfidence GVCF). The combined set of GVCF files were used when genotyping the entire set under joint calling mode (-T GenotypeGVCFs). Variant calls are available as a browser track at https://ricebase.org/jbrowse_ricebase/current/. Variants were merged with those derived from a previously sequenced set of ~3k rice varieties [22]; only intersecting variants were retained. 300 varieties from the 3k set, representing the 9 major subpopulations, were randomly down-sampled. This down-sampled set of 300 was combined with all US -developed varieties in this study. All single-nucleotide polymorphisms (SNPs) with a minor allele frequency less than .05 were removed. A reduced marker set was generated based on linkage disequilibria reduction as implemented in *plink* (—indep-pairwise 50 10 0.1).

### Population characterization

Population assignments and admixture estimation for US varieties was calculated using the 3k subset as the training set in *admixture*'s supervised mode (version 1.3.0,) with expected population number (K) equal to 9 [22]. To analyze changes through time, all US varieties (n = 166) were filtered to include only USA MidSouth sources (**Fig 3** and **S1 File**). Nira [PI305133] and Rexmont [GSOR 305081] were removed because of extreme divergence and poor genotyping, respectively, resulting in 101 accessions. Pruned-SNPs were further filtered from this set to remove any sites with heterozygosity >10%, as all varieties are inbred, and <20 lines without an allele call, resulting in 68,829 remaining sites. *Tassel* v5.5.50 [49] was used to create a kinship matrix from this marker set using the Centered-IBS method with maximum of 2 alleles [50].

### Calling SV genotypes

The insertions and deletions identified above were assessed for segregation across our entire resequenced population. Alignment files generated during SNP calling above were filtered to remove split and soft-clipped reads. Importantly, reads aligning to multiple locations were left in to preserve contiguity. SV-associated genomic intervals in the Nipponbare reference with >98% bases covered by one or more reads were considered support for the Nipponbare allele. If the Nipponbare interval was <20 bp (as in a CarGold insertion) then we required that at least one read span the entire interval and 5 bp flanking both sides. *bamtools/bedcov* commands used are provided (bedCovCommandForSegIndels.sh).

### Assessing allele frequency change through time

Both SV and SNP alleles were encoded as either 0 (ancestral) or 1 (derived) using information generated using the methodologies described above. The relationship between allelic state (y-values) and year-of-release (x-values) was fit to a linear model using the default *lm* function in R. Only polymorphisms with a minor allele frequency > 0.1 in Midsouthern material were used.

### Pedigree analysis

Twenty-seven trios were available for analysis within our dataset. Two trios, Palmyra and CI9701, were removed given the percentage of miscalls to segregating variants was >35% suggesting that pedigree records for these lines were incorrect. All SVs were evaluated based on the inheritance of each allele in the progeny. Only SVs segregating in 10 or more trios and, as above, that were derived in CarGold were kept (n = 553). Any SV for which a single allele was inherited in every progeny was removed, resulting in 548 total SVs. Though these SVs could be a result of extreme selection, they are more likely to be a result of miscalls in a common parent. Biased inheritance was evaluated as the number of derived alleles inherited divided by the sum of total segregating trios for the specific SV.

### Haplotype block analysis

For haplotype block assessment, the entire SNP set was filtered to MidSouth varieties (n = 101 described in "Population Characterization" method) and loci with MAF >10% and heterozygote calls in more than 10 varieties were removed, resulting in 810,808 sites. In addition, GSOR 305081 (Rexmont) was removed due to poor genotyping at these loci. *Tassel* v5.5.50 [49] was used to further downsample loci by restricting distance to neighboring SNP. Starting at a distance of 20K bp, we progressively reduced distance and assessed the number of resultant haplotype blocks as a proportion of total sites not in strong LD. This value plateaued at 3 kb (n = 59,575 SNPs). LD blocks were assessed using *snpldb* (v.1.2) [51], with MAF = 0.02 (at the haplotype level) and max window of 800 kbp. This 3 kb set and *h-scan* software (version 1.0: messerlab.org/resources/) was used to gauge the average length of pairwise identity tracts for each SNP position for each time period depicted in **Fig 5**.

## Supporting information

**S1 Fig. Sequencing coverage for single-copy regions plotted against time.**
(PDF)

**S2 Fig. Histogram of h-scan ratios for late versus early eras.**
(PDF)

**S3 Fig. Boxplot of h-scan ratios for putatively selected versus neutral loci.**
(PDF)

**S4 Fig. Dotplot of CarGold contigs (y-axis) relative to Nipponbare chromosomes (x-axis).**
(PNG)

**S5 Fig. Relationship between SV attributes and rate.** Insertions are light blue; deletions are black.
(PDF)

**S6 Fig. Feature density along chromosomes of different SV classes.** All chromosomes, regardless of length, shared the same x-axis.
(PDF)

**S7 Fig. Genomic profiles of HScan index in early versus late eras.**
(PDF)

**S8 Fig. Haplotype plots for all chromosomes.**
(PDF)

**S9 Fig. Relationship between year-of-release and genetic distance (centered IBS) to Nipponbare reference.**
(PDF)

**S10 Fig. Genotype recall and bias assessment.** A) Relationship between coverage and genome-wide genotyping as well as downsampled CarGold short-read sets, which are assumed to have zero true Nipponbare alleles based on assemblies. B) Linear relationship between coverage and false negative rate for each variant class.
(PDF)

**S11 Fig. Cumulative frequency plots for derived SV alleles in Midsouthern sample with a frequency between 0.1 and 0.9.** Each line represents the location of the SV relative to a gene.
(PDF)

**S12 Fig. Comparison of PacBio assembly of Carolina Gold (from this manuscript) with previous published Nanopore assembly of the same variety.** Chromosomes are shown in order 1 through 12. PacBio contigs from this study are on y-axis, Nanopore Carolina Gold Select [15] in first column and *Oryza sativa* var. indica in second column (ftp://ftp.gramene.org/pub/gramene/release-58/fasta/oryza_indica/dna/Oryza_indica.ASM465v1.dna_sm.toplevel.fa.gz).
(PNG)

**S13 Fig. Comparison of derived allele frequencies for indels in Midsouthern (n = 101) versus all other (n = 63) material resequenced as part of this study.** The reference containing the derived allele is indicated by color; line was fit using loess smoothing.
(PDF)

**S14 Fig. Trio call types for SVs in this study based on 27 available parent1-parent2-progeny trios available among resequenced lines in this study.** "ancInherit" and "derInherit" described cases in which the ancestral or derived allele was inherited, respectively. "misCall" describes cases in which the progeny allele is distinct from either parent allele.
(PDF)

**S15 Fig. Comparison of trio call types between SVs in this study and a subset of SNPs.** Note, 9 trios were removed from this comparison because they involved non-Midsouthern material and did not have complementary SNP data available. Axes represent the number of "misCalls", as defined in **S14 Fig**, as a percentage of segregating variants; SVs on y-axis, SNPs on x-axis.
(PDF)

**S16 Fig. Cumulative frequency distributions of inheritance patterns of derived alleles among SVs based on their position relative to protein coding structure.** Any SV that had <10 trios segregating was removed. Among these, any SV for which either allele was never inherited–suggesting false parent call—was also removed. Final counts for each class are

recorded in legend.
(PDF)

**S1 Table. Assembly statistics for Carolina Gold.**
(DOCX)

**S2 Table. Core gene content for Carolin Gold assembly and Nipponbare reference.**
(DOCX)

**S3 Table. GO analysis of genes associated with declining exonic SVs.**
(DOCX)

**S4 Table. GO analysis of genes associated with declining exonic and intronic SVs.**
(DOCX)

**S1 File. Master key with characteristics for all accessions sequenced as part of this study.**
(XLSX)

**S2 File. TE families and consensus sequences generated in this study.**
(TXT)

**S3 File. TE frequencies in indels.**
(XLSX)

**S4 File. SV information, excluding sequence.**
(XLSX)

**S5 File. Putatively selected loci based on sd1 threshold.**
(XLSX)

**S6 File. Agronomically important genes.**
(XLSX)

**S7 File. Genotyping matrix for all SVs in this study.** Accession names can be cross referenced using S1 File.
(TXT)

## Acknowledgments

Thanks to Amanda Hulse-Kemp and Aureliano Bombarely for review and comments. Thanks to Fanny Liu for laboratory assistance. Thanks to Brian Abernathy for data archiving. USDA is an equal opportunity provider and employer.

## Author Contributions

**Conceptualization:** Justin N. Vaughn, Jeremy D. Edwards, Doreen H. Ware, Anna M. McClung, Brian E. Scheffler.

**Data curation:** Justin N. Vaughn, Joshua C. Stein, Jeremy D. Edwards, Sheron A. Simpson, Ramey C. Youngblood, Jane Grimwood.

**Formal analysis:** Justin N. Vaughn, Walid Korani, Joshua C. Stein, Jeremy D. Edwards, Kapeel Chougule.

**Funding acquisition:** Daniel G. Peterson, Brian E. Scheffler.

**Methodology:** Justin N. Vaughn, Walid Korani, Joshua C. Stein, Sheron A. Simpson, Ramey C. Youngblood, Jane Grimwood.

**Project administration:** Brian E. Scheffler.

**Resources:** Jeremy D. Edwards, Doreen H. Ware, Anna M. McClung, Brian E. Scheffler.

**Software:** Justin N. Vaughn, Walid Korani.

**Supervision:** Justin N. Vaughn, Doreen H. Ware, Brian E. Scheffler.

**Validation:** Justin N. Vaughn.

**Visualization:** Justin N. Vaughn.

**Writing – original draft:** Justin N. Vaughn, Doreen H. Ware, Anna M. McClung.

**Writing – review & editing:** Justin N. Vaughn, Jeremy D. Edwards, Daniel G. Peterson, Doreen H. Ware, Anna M. McClung, Brian E. Scheffler.

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
