## [Decision Letter · Decision Letter 0]

28 Oct 2020

Dear Dr Vaughn,

Thank you very much for submitting your Research Article entitled 'Gene disruption by structural mutations drives selection in US rice breeding over the last century' to PLOS Genetics. Your manuscript was fully evaluated at the editorial level and by two independent peer reviewers. The reviewers appreciated the attention to an important problem, but raised some substantial concerns about the current manuscript. Based on the reviews, we will not be able to accept this version of the manuscript, but we would be willing to review a much-revised version. We cannot, of course, promise publication at that time.

If you decide to revise the manuscript for further consideration at PLOS Genetics, please aim to resubmit within the next 60 days, unless it will take extra time to address the concerns of the reviewers, in which case we would appreciate an expected resubmission date by email to plosgenetics@plos.org.

[LINK]

We are sorry that we cannot be more positive about your manuscript at this stage. Please do not hesitate to contact us if you have any concerns or questions.

Yours sincerely,

Gregory P. Copenhaver

Editor-in-Chief

PLOS Genetics

Reviewer's Responses to Questions

**Comments to the Authors:**

Reviewer #1: This paper describes the sequencing of Carolina Gold, a key rice variety in the US. The authors also do whole genome re-sequencing with 160+ varieties from the US breeding program. The authors use the generated sequence to examine the role of structural variants in the US breeding program, showing the pattern of SVs they observe and the shift over time.

This is a really nice paper that describes the dynamics of SVs during the breeding process. Normally papers that describe SVs in rice are not very compelling, but this one uses the data in an innovative way. It provides a glimpse of what happens in the course of breeding of a major crop, and is helped by the great amount of historical information (and seeds) of rice released in the US. In many ways, this could be a case study in historical genetics.

There are a few issues I think would help.

1. To what extent can they authors use the pedigree information and their ascertained SVs to even provide better resolution for the change in frequency over time; what SVs are generally lost?

2. It would be good to provide some examples of strong change in SV frequency and what kind of genes or regions these are found in.

3. Can the authors compare their SVs in this dataset with the frequency found in the worldwide 3,000 genome collection? The US breeding population is essentially a relatively closed gene pool, so it would be interesting to see if SVs of high frequency in the worldwide japonica collection are low in the US and vice-versa.

4. I think another Carolina Gold sequence was release last year (published also in PLos Genetics), but the authors do not refer to it at all. It would be good to refer to this paper and possibly make some comparison of the two genomes, given that the previously published paper used nanopore sequencing.

Reviewer #2: The study reports the detection of structural mutations and their population dynamics during modern US rice breeding. The purpose of the study is to detect structural mutations with >= 50 bp, and to characterize how these structural mutations were changed during modern US rice breeding. To address the questions, the study generated a de novo genome assembly of a Tropical japonica cv. Carolina Gold with PacBio long-read sequencing technology. Comparing the assembled genome sequence with Nipponbare reference genome with O.glaberrima as the outgroup yielded numerous of structural mutations. Tracing the variation of these structural mutations in 101 rice varieties bred at different eras in last century in the US showed they were more intensively selected than any other genomic variations during US rice breeding. To my opinion, recent rice genome study mainly focused on Asia landraces and varieties, the genetic architecture of US rice varieties remains unknown. This study could advance our understanding of genetic improvement during modern US rice breeding and provide a valuable resource for further comprehensive rice genome study. It is worthy of publication once following issues be addressed.

Major comments:

Rice recent functional genome studies have cloned many key genes for grain size and shape, eg, GW2, GS3, GS5, qGL3, G1, GS6 and qGW8, the authors should evaluate the genotype change of these genes to elucidate the genetic basis underlying the grain shape change during US rice breeding, and evaluate the potential impact of structural mutations on these genes.

Structural mutations strongly correlated with the population structure. In the study, unevenly population structure with genetic introgression was observed during modern US rice breeding. The effect of population structure change and early genetic introgression on the purge of structural mutations should be evaluated before drawing the conclusion “Structural mutations are being purged from the germplasm”.

Recent rice genome studies have identified large number of key genes were selected during modern breeding (eg. Xie et al, Breeding signatures of rice improvement revealed by a genomic variation map from a large germplasm collection; Lin et al, Divergent selection and genetic introgression shape the genome landscape of heterosis in hybrid rice; Li et al, Analysis of genetic architecture and favorable allele usage of agronomic traits in a large collection of Chinese rice accessions). It is interesting that structural mutations were purged during US rice breeding. The functional effect of structural mutations on important agronomic important genes, or the relationship between structural mutations and recently reported selected genes would be helpful for explaining why structural mutations were selected during modern breeding.

Minor comments:

Lacking of reference or description to O.glaberrima genome in line 127.

As the genome assembly quality will affect the result of structural detection, the authors should provide a brief description about the quality of O.glaberrima genome.

Ambiguous words like “likely” or “roughly” should not be used to describe the results, eg at line176 and line 251, explicit number should be investigated across the manuscript.

More detailed information about the trait improvement during US rice breeding, eg. the quality, resistance and yield change, should be provided to help understanding the characteristic of US breeding improvement.

**Have all data underlying the figures and results presented in the manuscript been provided?**

Reviewer #1: Yes

Reviewer #2: Yes

PLOS authors have the option to publish the peer review history of their article (what does this mean?). If published, this will include your full peer review and any attached files.

Reviewer #1: No

Reviewer #2: No

---

## [Decision Letter · Decision Letter 1]

28 Jan 2021

Dear Dr Vaughn,

We are pleased to inform you that your manuscript entitled "Gene disruption by structural mutations drives selection in US rice breeding over the last century" has been editorially accepted for publication in PLOS Genetics. Congratulations!

Yours sincerely,

Gregory P. Copenhaver, Ph.D.

Editor-in-Chief

PLOS Genetics

Comments from the reviewers (if applicable):

Reviewer's Responses to Questions

**Comments to the Authors:**

Reviewer #1: The issues raised in the previous review have been addressed

**Have all data underlying the figures and results presented in the manuscript been provided?**

Reviewer #1: Yes

PLOS authors have the option to publish the peer review history of their article (what does this mean?). If published, this will include your full peer review and any attached files.

Reviewer #1: No

**Data Deposition**

http://datadryad.org/submit?journalID=pgenetics&manu=PGENETICS-D-20-01339R1

**Press Queries**

---

## [Editor Report · Acceptance letter]

22 Feb 2021

PGENETICS-D-20-01339R1 

Gene disruption by structural mutations drives selection in US rice breeding over the last century 

Dear Dr Vaughn, 

We are pleased to inform you that your manuscript entitled "Gene disruption by structural mutations drives selection in US rice breeding over the last century" has been formally accepted for publication in PLOS Genetics! Your manuscript is now with our production department and you will be notified of the publication date in due course.

With kind regards,

Alice Ellingham

PLOS Genetics

On behalf of:
